# COVID-19 Pandemic Highlights the Importance of Inclusive Leadership in Egyptian Hospitals to Improve Nurses' Psychological Distress

**Eman Salman Taie [1],\* and Mohamed A. Zoromba [2]**

1    Nursing Administration, Faculty of Nursing, Helwan University, Cairo 11795, Egypt
2    Faculty of Nursing, Mansoura University, Mansoura 35516, Egypt
\*    Correspondence: dr_emys@hotamil.com or eman.salman@nursing.helwan.edu.eg

**Abstract: Background**: The pandemic has highlighted the crucial role of nurses in health services. Nurses work at the forefront of the healthcare system, provide infection control training, and help reduce the spread of misinformation about the pandemic. Inclusive leaders create psychological safety that improves motivation and boosts job performance. **Aim:** To explore the effect of nurse managers' inclusive leadership style on nurses' psychological distress during the COVID-19 pandemic in Egyptian hospitals. **Method:** This was a quasi-experimental study. The study subjects consisted of two groups: managers (171) and staff nurses (1573). The study was conducted in four hospitals (one university hospital, one private hospital, one therapeutic institution, and one health insurance hospital). Three tools were used for collecting data (Inclusive Leader Questionnaire, Nurse Managers' Knowledge about Inclusive Leadership, and Kessler Psychological Distress Scale). **Results:** In total, 67.25% of staff nurses perceived their nurse managers as poor inclusive leaders, and only 12.86% perceived them as good inclusive leaders. Regarding nurse managers' knowledge about inclusive leadership, 76.023% had unsatisfactory knowledge levels before awareness sessions, and only 7.017% had a satisfactory level compared to after awareness sessions, when the majority of them had satisfactory knowledge levels. Furthermore, before awareness sessions, staff nurses experienced mild to moderate psychological distress during the COVID-19 pandemic, and only 8.2% were severe. After awareness sessions, 58.55% of them experienced mild psychological distress, and others became well (25.8%). **Conclusions:** Nurse managers lack knowledge about inclusive leadership before conducting awareness sessions. The majority of staff nurses perceived their nurse managers as poor inclusive leaders. Satisfactory knowledge levels among nurse managers after awareness sessions improved nurses' psychological distress. Finally, there were strong, statistically significant positive correlations between inclusive leadership and nurses' psychological distress.

**Keywords:** inclusive leader; psychological distress; COVID-19 pandemic

## 1. Introduction

The world today is experiencing a variety of events, crises, and illnesses. These crises threaten physical and spiritual human life despite scientific advances. Humans have not yet been able to fully overcome these crises. Traumatic events such as the coronavirus disease (COVID-19) outbreak were first reported in December 2019 in Wuhan City, China. The COVID-19 outbreak has changed employees' cognitive patterns and their perception of the world as a safe place and shattered the assumption that a workplace is a safe place. Hospitals and medical staff are one of the first and most important factors in crisis management due to their professional nature [1–3].

Healthcare workers around the world have been greatly impacted by COVID-19, since they work on the front lines of the pandemic [4]. A perceived gap between the external demands placed on an individual and their available coping resources leads to stress.

There are various stressors in the workplace and their impact on the psychological state of healthcare professionals during the COVID-19 pandemic is significant. Some of these stressors include but are not limited to fear of transmission, a workload that interferes with private life, uncertainty/lack of knowledge, and group anxiety [5,6]. To cope with stressors in the workplace, healthcare professionals need field-specific psychosocial support in and out of the workplace to reduce stress responses and protect their mental health, well-being, and functioning [7,8]. Healthcare workers are at high risk for mental health problems, such as depression, anxiety, distress, and insomnia, which have increased dramatically during the pandemic. Healthcare professionals have been the most important resource in saving lives and limiting the impact of the pandemic, so their health is very important to the quality of service they provide [9–11].

Many factors play a role in the psychological distress nurses may face due to the spread of COVID-19. These include high rates of infectivity and mortality, a long incubation period for the virus, retention of information and uncertain information about the mode of transmission, and necessary precautions [12,13]. Another factor that could weigh on health workers is isolation and quarantine. Nurses have been exposed to life-threatening occupational hazards throughout the COVID-19 pandemic; they face loneliness while remaining isolated and confined due to the highly contagious nature of the virus. Some even abstain from consuming liquids to keep from taking restroom breaks because it may require converting protective gear, and a few shave their heads to lessen infection from sweating. Their correct intellectual fitness is critical to maximize their potential to take care of sufferers of infectious diseases [14,15].

Due to the emergence of crises and changing conditions, management in the healthcare field faces various challenges. The crisis also challenged nurse leaders to develop and implement novel care-delivery plans while preventing disease transmission to patients and staff. The COVID-19 pandemic has altered the way health professionals work and accelerated changes that may previously have taken ten years. Leaders are being challenged to adapt at the same pace [16–19]. Leaders must prioritize inclusion now more than ever. Organizations are much more likely to be innovative in the face of this crisis if they seek input from a diverse group of employees who approach issues from different angles. At the same time, employees from historically underrepresented groups may feel less comfortable talking [20–22].

In the current situation, inclusion is particularly important to help fully realize a sense of teamwork, safety culture, and belonging. Inclusion is the full acceptance and integration of all employees, regardless of their diversity [21,23,24]. For employees, it proactively leads to a sense of belonging, commitment, progress, and full participation within the organization. Inclusive leadership refers to leaders who demonstrate visibility, accessibility, and availability when interacting with their subordinates. Inclusive leadership is an approach where all voices are heard, and it means being curious and aware of differences at the same time. This includes establishing high levels of trust, articulating a clear purpose, and exploring different perspectives to make better decisions. Inclusive leaders create psychological safety that improves motivation and boosts job performance. In addition, inclusive leadership goes beyond traditional leadership to foster flourishing engagement on a deeper, more personal level; it values authenticity and wholeness. Inclusive leaders get the most out of all their employees, helping their organizations succeed in today's complex and diverse domestic and global environment. Through their adaptability, relationship building, and talent development skills, inclusive leaders are empowered to increase performance and innovation [7,20,25]. The pandemic has highlighted the crucial role of nurses in health services. Nurses work at the forefront of the healthcare system, provide infection control training, and help reduce the spread of misinformation about the pandemic. The need to be aware of the acute and long-term mental health consequences for healthcare professionals is emerging [13,19].

The impact of COVID-19 has been felt around the world, with large organizations quickly turning their attention to security, financial, or economic impacts. Therefore,

healthcare organizations must be able to respond to the immediate COVID-19 crisis and its long-term effects. Inclusive leaders strive to minimize these differences between themselves and their subordinates and to ensure that employees are recognized for their contributions, regardless of their rank in the workplace. The benefits of having inclusive leaders who place a high value on the diversity and authenticity of their workforce include the creation of a growing and engaged team, encouragement of collaborative relationships and respect, and the development of a dynamic and innovative environment. In turn, this will increase productivity and earnings, enhance employer branding, improve employee empowerment, and promote engagement with and ownership of organizational success [3,17,19].

The situation in Egypt is getting worse, with an increase in the number of cases and the number of deaths and the possibility that the actual figures are higher than those reported. Furthermore, with public health spending in Egypt accounting for only 5.3% of gross domestic product (GDP), these limited resources, combined with the growing number of cases, place a tremendous burden on the medical staff in the health sector. The Egyptian Ministry of Health on March 31 announced two hotlines assigned to the General Secretary of Mental Health to provide psychological support to people (including healthcare providers) during the time of the pandemic outbreak of COVID-19 [26–30].

*Aim of the Study*

The present study aims to explore the effect of nurse managers' inclusive leadership style on nurses' psychological distress during the COVID-19 pandemic in Egyptian hospitals through:

1. An assessment of inclusive leadership for nurse managers as perceived by staff nurses in selected hospitals.
2. An assessment of nurse managers' knowledge about inclusive leadership.
3. Raising nurse managers' awareness about inclusive leadership.
4. The identification of staff nurses' psychological distress levels before and after awareness sessions.
5. An investigation of the correlation between inclusive leadership and staff nurses' psychological distress.

It is hypothesized that most nurse managers lack knowledge about inclusive leadership. Additionally, there is a positive correlation between nurse managers' inclusive leadership style and staff nurses' psychological distress levels.

## 2. Materials and Methods

### 2.1. Research Design

This was a quasi-experimental study.

### 2.2. Study Setting

The study was conducted in one university hospital (Ain Shams Specialized Hospital), a therapeutic institution in Cairo under the Ministry of Health and Population (MOHP) (Dar el-Shefaa Hospital in Cairo), one health insurance hospital (El-Nasr Hospital, affiliated with the health insurance sector in Helwan governorate), and one private hospital (El-Salam International Hospital).

### 2.3. Participants

The study subjects consisted of two groups (Table 1):

*1st group:* This group consisted of all nurse managers in the selected hospitals, including all categories as head nurses, supervisors, and nursing directors. All available nurse managers of both genders with at least two years' experience were included in the study (*n* = 171). Those who attended any previous training about inclusive leadership were excluded.

*2nd group:* This group composed of all staff nurses who are working in the previously mentioned hospitals at the time of data collection and agreed to participate in the study.

Participants included nurses of both genders who had at least one year of experience in the hospital before the onset of the COVID-19 pandemic.

**Table 1.** Distribution of study subjects according to hospitals.

| Hospitals | Nurse Managers (*n* = 171) | | Staff Nurses (*n* = 1573) | |
|---|---|---|---|---|
| | No. | % | No. | % |
| **University** | | | | |
| Ain Shams Specialized Hospital | 75 | 43.859 | 687 | 43.67 |
| **Therapeutic Institution** | | | | |
| Dar el-Shefaa | 41 | 23.976 | 292 | 18.56 |
| **Health Insurance** | | | | |
| El-Nasr | 30 | 17.543 | 236 | 15 |
| **Private** | | | | |
| El-Salam International | 25 | 14.619 | 358 | 22.759 |

*2.4. Study Instruments*

2.4.1. Inclusive Leader Questionnaire Format

This questionnaire was designed by [28] and modified by the researchers.

It was composed of two parts:

*Part I: Personal data of staff nurses:* This part included age, gender, level of education in nursing, years of experience, etc.

*Part II: Inclusive leader questionnaire:* This was a self-administered questionnaire by staff nurses. It consisted of 40 items divided into four main dimensions: *Dimension 1:* 10 items, including providing equal opportunity and fair treatment to all work unit members; *Dimension 2:* 18 items, including encouraging the integration of and synergy among all work unit members; *Dimension 3:* 9 items, including directly addressing work unit members' fundamental needs for uniqueness, authenticity, and belongingness; and *Dimension 4*: 3 items, including implementing organizational diversity and inclusion-related policies and programs in the work unit.

*Scoring system:* The total score was 200. Staff nurses' responses were measured on a five-point Likert scale: 1 = Almost never, 2 = Seldom, 3 = Sometimes, 4 = Often, 5 = Almost always. The inclusive leadership measure was considered:

- Good >80% (>160);
- Moderate 70% =< 80% (140 =< 160);
- Poor 70% (<140).

Validity and reliability testing for this study showed a Cronbach's alpha coefficient of the instrument at 0.82 for the study sample. The questionnaire had high construct validity (with a part–whole correlation of 0.90).

2.4.2. Nurse Managers' Knowledge about Inclusive Leadership Questionnaire Format

This tool was designed by the researcher after reviewing the relevant literature [19,24]. It was a self-administered questionnaire used to assess the selected nurse managers' knowledge about inclusive leadership. It included questions regarding topics such as the definition, traits, competencies, roles, and importance of an inclusive leader. The Cronbach alpha coefficient of the questionnaire was 0.88 for the study sample. The questionnaire had high construct validity (with a part–whole correlation of 0.91).

*Scoring system:* The total score was 10. Nurse managers' responses were measured on a two-point scale: 1 = right and 0 = wrong. Knowledge level was considered:

- Satisfactory (9–10);
- Moderate (7–8);

- Unsatisfactory (1–6).

### 2.4.3. Kessler Psychological Distress Scale (K10)

This tool was developed by Kessler et.al. 2003 [31]. The Kessler Psychological Distress Scale is a simple measure used to assess staff nurses' psychological distress during the COVID-19 pandemic. The K10 scale involves 10 questions about emotional states, each with a five-level response scale. This scale was adopted by the researcher. It was a self-administered questionnaire.

*Scoring system:* Each item was scored from 1 'none of the time' to 5 'all of the time'. Scores of the 10 items were then summed, yielding a minimum score of 10 and a maximum score of 50. Low scores indicated low levels of psychological distress, and high scores indicated high levels of psychological distress.

Interpretation of scores

*K10 Score:* Likelihood of having a mental disorder (psychological distress).

10–19 Likely to be well;

20–24 Likely to have a mild disorder;

25–29 Likely to have a moderate disorder;

30–50 Likely to have a severe disorder.

### 2.5. Procedure

To carry out the study in the predetermined hospitals, letters containing the aim of the study were directed from the researcher's faculty of Nursing to each hospital general manager to obtain their permission and help to conduct the study in their facility. The researchers obtained approval from the scientific research ethical board for the faculty of Nursing. Then, informed consent was obtained from all subjects involved in the study after explaining the purpose and method of data collection to them. Confidentiality, anonymity, and the right to withdraw from the study at any time were guaranteed.

The researchers randomly selected 17 nurse managers from all of the study's hospital settings. In addition, 200 staff nurses were selected for the pilot study. Based on the pilot study, no modifications were required. Therefore, all of these subjects were included in the main study sample.

This study was executed from the beginning of August 2020 to the end of March 2022. The researchers started to assess inclusive leadership for nurse managers as perceived by staff nurses. The time needed by staff nurses to fill out the Inclusive Leader Questionnaire ranged 20–25 min. Then, the researchers assessed nurse managers' knowledge about an inclusive leadership style. The time needed to fill out the questionnaire ranged 10–15 min. Finally, the researchers assessed staff nurses' psychological distress levels in the selected hospitals using the Kessler Psychological Distress Scale, for which the time required ranged 10–15 min and which lasted for four months. Then, the awareness sessions were conducted for managers for five hours per day; every awareness session included between 18 and 20 attendees. This lasted for five months. A pause occurred due to fourth and fifth waves of COVID-19. After awareness sessions, nurse managers were assessed again for their knowledge about inclusive leadership styles, and staff nurses were assessed again for their psychological distress levels; this lasted for two months.

### 2.6. Statistical Analysis

The SPSS version 25 statistical software package was used for data analysis. Chi-squared ($\chi^2$) was used for comparison between qualitative variables. The probability of error at 0.05 was considered significant, while at 0.01 and 0.001, it was considered highly significant. A Pearson correlation analysis was used to assess the inter-relationships among quantitative variables.

## 3. Results

It can be observed from Figure 1 that more than two-thirds of staff nurses (67.25%) perceived their nurse manager as a poor inclusive leader. Meanwhile, only 12.86% perceived them as a good inclusive leader.

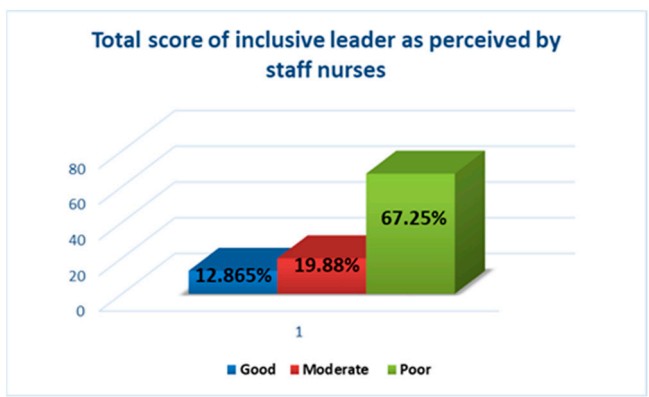

**Figure 1.** Distribution of total score of inclusive leaders, as perceived by staff nurses (*n* = 171).

Table 2 describes the personal data of the studied nurse managers. It illustrates that the highest percentage (71.34%) of the participants were aged between 35 and 45, while only 7% were over 45 years old. Regarding their educational level in nursing, just over half (55.55%) held a Bachelor's degree in nursing, while only 8.77% had a PhD in nursing. Moreover, as demonstrated in the following table, about half (50.8%) had between >10 and <15 years of experience, as compared to only 8.18% with less than 5 years of work experience.

**Table 2.** Personal data of studied nurse managers (*n* = 171).

| | Nurse Managers (*n* = 171) | |
|---|---|---|
| **Personal Data** | **N** | **%** |
| **Age (years)** | | |
| <25 | 0 | 0.0 |
| 25 =< 35 | 37 | 21.63 |
| 35 =< 45 | 122 | 71.34 |
| >45 | 12 | 7 |
| **Educational level in nursing** | | |
| Bachelor degree | 95 | 55.55 |
| Master degree | 61 | 35.67 |
| Doctorate degree | 15 | 8.77 |
| **Years of experience** | | |
| <5 years | 14 | 8.18 |
| 5 =< 10 years | 39 | 22.8 |
| >=10 =< 15 years | 87 | 50.8 |
| =<15 years | 31 | 18.12 |

It can be seen in Table 3 that more than two thirds (66.43%) of the participating staff nurses were aged between 35 and 45 years old; meanwhile, only 2.41% were under 25 years old. In regard to their educational level in nursing, more than half (55.43%) held a Bachelor's degree and one third (33.63%) had graduated from a technical nursing institute. Only 10.93% had a Master's degree in nursing and none of the participants held a PhD in nursing. Furthermore, as evident from the table, almost two thirds (62.55%) had between >10 and <15 years of experience, and only 1.589% had less than 5 years of experience.

**Table 3.** Personal data of studied staff nurses (*n* = 1573).

| | Staff Nurses (*n* = 1573) | |
|---|---|---|
| **Personal Data** | **N** | **%** |
| **Age (years)** | | |
| <25 | 38 | 2.41 |
| 25 =< 35 | 408 | 25.93 |
| 35 =< 45 | 1045 | 66.43 |
| >45 | 82 | 5.21 |
| **Educational level in nursing** | | |
| Technical Nursing Institute | 529 | 33.63 |
| Bachelor degree | 872 | 55.43 |
| Master degree | 172 | 10.93 |
| Doctorate degree | 0 | 0 |
| **Years of experience** | | |
| <5 years | 25 | 1.589 |
| 5 =< 10 years | 378 | 24.03 |
| >=10 =< 15 years | 984 | 62.55 |
| =<15 years | 186 | 11.82 |

Table 4 demonstrates that more than three quarters of the nurse managers (76.023%) had unsatisfactory knowledge about inclusive leadership, and only 7.017% had satisfactory knowledge levels before the awareness sessions. However, the post-awareness session data shows that the majority (90.64%) had satisfactory knowledge levels and only two managers (1.169%) were had unsatisfactory knowledge levels. There was a high significant (*p* < 0.0001) difference between nurse managers' knowledge level about inclusive leadership before and after awareness sessions.

**Table 4.** Percentage distribution of nurse managers' knowledge about inclusive leadership before and after awareness sessions (*n* = 171).

| Knowledge | Before Awareness Sessions | | After Awareness Sessions | | *p*-Value |
|---|---|---|---|---|---|
| | **N** | **%** | **N** | **%** | |
| **Satisfactory** | 12 | 7.017 | 155 | 90.64 | |
| **Moderate** | 29 | 16.959 | 14 | 8.187 | ** 0.0001 |
| **Unsatisfactory** | 130 | 76.023 | 2 | 1.169 | |

** Highly significant *p* < 0.01.

Table 5 shows that before the awareness session, 51.3% and 25.49% of the respondents experienced mild to moderate psychological distress during the COVID-19 pandemic, while only 8.2% suffered severe disorders and the rest (15%) were well. Comparing this to the post-awareness session data, 58.55% of the participants experienced mild psychological distress and a higher percentage were well (25.8%). Meanwhile, only 13.159% and 2.479% of the respondents experienced moderate to severe psychological distress. There was a high significant (*p* < 0.0001) difference between staff nurses' psychological distress during COVID-19 before and after the awareness sessions.

Table 6 highlights strong statistically significant positive correlations between the two parameters (r = 0.89). There was a high significant (*p* < 0.001) difference between them.

**Table 5.** Percentage distribution of staff nurses' psychological distress during COVID-19 pandemic before and after awareness sessions (*n* = 1573).

| Psychological Distress | Before Awareness Sessions | | After Awareness Sessions | | *p*-Value |
|---|---|---|---|---|---|
| | **N** | **%** | **N** | **%** | |
| **Well** | 236 | 15 | 406 | 25.8 | |
| **Mild disorder** | 807 | 51.3 | 921 | 58.55 | ** 0.0001 |
| **Moderate disorder** | 401 | 25.49 | 207 | 13.159 | |
| **Severe disorder** | 129 | 8.2 | 39 | 2.479 | |

** Highly significant *p* < 0.01.

**Table 6.** Correlation between inclusive leadership as perceived by staff nurses and their psychological distress.

| Variables | Pearson Correlation Coefficient | *p*-Value |
|---|---|---|
| Inclusive leadership | | |
| Psychological distress | 0.89 * | ** 0.001 |

* Significant *p* < 0.05; ** Highly significant *p* < 0.01.

## 4. Discussion

There is no doubt that major traumatic events, such as the COVID-19 pandemic, have highlighted the vital role of nurses in health services; therefore, the quality of provided care for infected individuals was affected by the state of the mental well-being of nurses, which demanded research that anticipates ways to enhance the mental health of nurses and alleviate their psychological distress. Reacting to these demands for research on the effect of a leadership style on psychological distress during this crisis, the current paper contributes to the literature on exploring the effect of nurse managers' inclusive leadership style on nurses' psychological distress during the identified pandemic. However, the current study revealed that a prominent percentage of staff nurses perceived their nurse manager as a poor inclusive leader. These findings were inconsistent with [13,22], who asserted that inclusive leaders are folks who seek out and weigh diverse perspectives, create an experience of belonging, and build deep alignment on a clear purpose. This agrees with [21,24], who found that inclusive leaders are not just the greatest at developing high-performing teams, but they are also the best at engaging and encouraging people to collaborate successfully in a difficult environment, for instance, over physical distance and video displays.

In the same vein, Refs. [7,19] emphasized that inclusive leaders help organizations to be resilient and to persevere through the immediate of crisis COVID-19 and its long-term effects. On the other side, Refs. [18,19] asserted that before this crisis, leadership groups that did not replicate the demographic realities of today's markets and talent pools may be inadvertently creating risk by being out of synchronization and may be unable to cope quickly enough with today's realities and crises. They added that leaders face complicated challenges that may need the most effective thinking to come up with solutions.

When employees feel comfortable asking for help, sharing recommendations informally, or challenging the status quo without the concern of terrible social consequences, corporations have greater possibilities to innovate quickly, reap the benefits of diversity, and adapt nicely to full utilize abilities that have only grown in importance throughout the COVID-19 crisis. The results of the present study showed that managers lack knowledge about inclusive leadership. These results were in contrast with [23], who asserted that inclusive leaders commit to making range and inclusion organizational priorities with the aid of engaging all tiers of management and teams of workers in the organization, growing formal strategies and buildings to create a high-quality work environment, and engaging employees from more than a few backgrounds. This was in harmony with [2,3], who found

that inclusive leaders provide not simply intellectual support but also join with subordinates at an emotional level. This continues in such a work context that helps personnel achieve a higher degree of psychological safety and motivates them in the direction of proactive behavior.

Interestingly, the current study findings revealed that staff nurses exhibited significant differences regarding their psychological distress before and after awareness sessions, as more than three-quarters of them experienced mild to moderate psychological distress during the COVID-19 pandemic before awareness sessions, compared to the majority of them who experienced well to mild psychological distress after awareness sessions. This was supported by [32], who conducted a cross-sectional survey that enrolled 502 participants and revealed a high prevalence of mental health symptoms among healthcare workers treating COVID-19 patients in Egypt. Overall, 77.3%, 69.5%, 79.3%, and 83.1% of all participants reported symptoms of anxiety, insomnia, depression, and stress, respectively. These results may reflect the difficult situation facing Egypt's frontline healthcare workers. This may be related to Egypt's limited number of nurses and physicians, reported by the World Bank, at 1.9 nurses per 1000 people and 0.5 doctors per 1000 people, which is lower than in most European and Asian countries, including China. Add to that the presence of a low number of hospital beds available (1.6 per 1000 population) [19]. These findings were in the same line with [33], who found in a study on Egyptian nurses that a quarter of the nurses studied had moderate levels of depression and anxiety, and more than one in ten had moderate levels of stress.

These findings are in harmony with the literature that established cumulative evidence on the effect of training nurse managers about varied positive leadership styles and their outcomes on enhancing their staff's attitude, thinking, and behavior. A systematic review conducted by [34–37] concluded that leadership interventions had a beneficial effect on the leadership behaviors of participants based on both subjective and objective changes in behavior in addition to focus on individual skill development. Leadership training plays a significant part in providing actual and effective care and leads to positive results for patients, health professionals, and the work environment.

Congruently, a study in Vietnam reported that inclusive leadership had a positive relationship with the well-being and innovative behavior of employees [17]. In addition, Ref. [37] in Egypt, Ref. [18] in Pakistan, and [38,39] in China confirmed the effect of inclusive leadership on innovative work behavior. Moreover, inclusive leadership is associated with nurses' psychological safety, speaking up, error-reporting intention, perceived organizational support, and employee work engagement [19,38,40]. Furthermore, inclusive leadership indicates morality in listening to all ideas, showing attendance of managers as employees meet challenges, and having the talent to discuss critical and vital actions with employees. In [41], researchers revealed that inclusive head nurses inspire and respect staff to accomplish challenging and hard goals, recognize and appraise their efforts and accomplishments to realize those identified goals, and show a reactive attitude where leaders react clearly and efficiently to personnel complications. Likewise, they delegated power to their staff, with elevated autonomy to do their work duties with their own approach. Moreover, they appreciated the existence of staff in a work setting.

In the current study, strong, statistically significant positive correlations were revealed among the two studied variables: inclusive leadership as perceived by staff nurses and their psychological distress. The results of this study are in the same line as some previous studies in which head nurses were shown to reveal very low levels of inclusive leadership [39]; in addition, inclusive leadership has a negative correlative relationship with psychological distress [36,39].

The current findings along with previous studies show that enhancing inclusive leadership works as a sustainable mechanism to decrease psychological distress throughout pandemics [42]. Inclusive leaders should work to produce a more open and engaging environment for nurses. This, in turn, supports enhancing nurses' focus and engagement while decreasing their psychological distress. Accordingly, it can operationalize the delivery

of mental health support in work settings. Harmoniously, Ref. [39] stated that head nurses are critically required to review their inclusive leadership skills. All of the previous researchers assert the need for positive leadership styles, such as inclusive leadership, to bring healthcare workers together through caring and sharing behavior. This makes subordinates feel psychologically safe and gives them the mental strength to continue fighting illnesses such as COVID-19 that cause public health emergencies and trauma.

*Limitations of the Study*

COVID-19 waves (fourth and fifth) paused study data collection procedure.

## 5. Conclusions

This study contributes to shedding new light on the effect of nurse managers' inclusive leadership style on nurses' psychological distress during the COVID-19 pandemic in Egyptian hospitals. The study assessed inclusive leadership for nurse managers as perceived by staff nurses. About two-thirds of them perceived their nurse manager as a poor inclusive leader, and only 12.86% perceived their nurse managers as a good inclusive leader. The study also assessed managers' knowledge about inclusive leadership. The majority of them had unsatisfactory knowledge about inclusive leadership, and only 7.017% of them had satisfactory knowledge levels before awareness sessions. Awareness sessions were conducted. After the awareness sessions, the majority of them had satisfactory knowledge, and only two managers had unsatisfactory knowledge levels. The study also analyzed staff nurses' psychological distress levels before and after awareness sessions. The study found that 51.3%, 25.49%, and 8.2% of the respondents experienced mild, moderate, and severe psychological distress during COVID-19, respectively, and 15% were well before awareness sessions. This indicates the successful effect of the awareness sessions. Finally, the study highlighted strong, statistically significant positive correlations between inclusive leadership and staff nurses' mental health.

## 6. Recommendations

1. Nursing curricula should introduce different leadership styles that suit the current COVID-19 pandemic, such as inclusive leadership.
2. Healthcare organizations should create an inclusive environment and open and accessible leadership as COVID-19 changes the workforce.
3. Organizational support should include guarantees for the healthcare workforce who fall ill, through medical and financial support for them and their families and protection from threats of neglect.
4. Encourage managers to attend training programs, conferences, and seminars about inclusive leadership styles.
5. Future research about inclusive leaders and psychological distress among nurses during COVID-19 in different Egyptian hospitals will provide more rigorous results.

**Author Contributions:** Conceptualization, E.S.T.; methodology E.S.T.; software, E.S.T. and M.A.Z.; validation, E.S.T.; formal analysis, E.S.T.; investigation E.S.T.; resources, E.S.T. and M.A.Z.; data curation, E.S.T.; writing—original draft E.S.T. and M.A.Z. All authors have read and agreed to the published version of the manuscript.

**Funding:** This research received no external funding.

**Institutional Review Board Statement:** The study was approved by scientific research ethical committee at faculty of nursing, Helwan University.

**Informed Consent Statement:** Informed consent was obtained from all subjects involved in the study.

**Data Availability Statement:** Not applicated.

**Conflicts of Interest:** The author declares no conflict of interest.

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
