# Peer review of "COVID-19 Pandemic Highlights the Importance of Inclusive Leadership in Egyptian Hospitals to Improve Nurses’ Psychological Distress"

_psych, doi:10.3390/psych4030041_

Round 1

Reviewer 1 Report

Thank you for giving me the opportunity to review this manuscript. The subject is very relevant and current. The health crisis triggered by COVID-19 has been a heavy overload for health professionals, especially nurses. This extraordinary situation has had an impact on their health causing strong psychological discomfort. The measures to deal with this problem must come from various levels, including from the leaders by exercising inclusive leadership that strengthens the team, which is particularly vulnerable due to the pandemic.

Introduction

I would suggest expanding the importance of inclusive leadership in times of COVID-19 pandemic, this is to say, the need of inclusive leadership, the benefits…

Line 33: “a variety of events, crises and illnesses” is too ambiguous, I suggest rephrasing this line.

References tend to be placed at the end of the paragraph, instead of at the end of the idea or argument. I recommend placing the reference just at the end of the argument which is supporting.

Line 39: which epidemiological work? Please, provide references.

There are plenty of studies focused on healthcare workers’ psychological distress. I suggest expanding the introduction in this line. I recommend including the references:

Gómez-Salgado, J., Domínguez-Salas, S., Romero-Martín, M., Romero, A., Coronado-Vázquez, V., & Ruiz-Frutos, C. (2021). Work engagement and psychological distress of health professionals during the COVID-19 pandemic. Journal of nursing management, 29(5), 1016–1025. https://doi.org/10.1111/jonm.13239

Domínguez-Salas, S., Gómez-Salgado, J., Guillén-Gestoso, C., Romero-Martín, M., Ortega-Moreno, M., & Ruiz-Frutos, C. (2021). Health care workers' protection and psychological safety during the COVID-19 pandemic in Spain. Journal of nursing management, 29(7), 1924–1933. https://doi.org/10.1111/jonm.13331

Ruiz-Frutos, C., Ortega-Moreno, M., Soriano-Tarín, G., Romero-Martín, M., Allande-Cussó, R., Cabanillas-Moruno, J. L., & Gómez-Salgado, J. (2021). Psychological Distress Among Occupational Health Professionals During Coronavirus Disease 2019 Pandemic in Spain: Description and Effect of Work Engagement and Work Environment. Frontiers in psychology, 12, 765169. https://doi.org/10.3389/fpsyg.2021.765169

Line 64: Please, eliminate the commas or cite the quotation.

I suggest the introduction in one section, eliminate the epigraphs 1.1, 1.2 and 1.3 and include them all in one section linked to each other.

Research hypotheses: you state five objectives but only two hypotheses.

Materials and methods

Subsections should start with number 2.

Study design: I would say this is a descriptive, cross sectional study.

Study settings: Please, describe briefly the four hospitals.

Subjects: In this section you describe the participants. First you should describe the study population, number and characteristics of the nurse managers and the staff nurses in each participating hospital. This will allow knowing if the sample is representative.

Also, you need to describe how the participants were selected, the sample procedure.

Line 120: what do you mean with “available”? I think this is not the right word here.

Line 121: Why did you exclude nurses with previous training in inclusive leadership?

Please provide sociodemographic data regarding the participants.

Line 138: I think it should be part II.

Line 134 and the rest of the document: reference 25 is not correct, it should say the name of the author, and in case of multiple authors, the name of the first author followed by et al. Please, correct this in the whole document.

Line 138: please eliminate the parenthesis for the figure, It consisted of 40 items…The same in line 144, 158, 179, 180 and 182.

When describing the scoring system of the assessing tools, eliminate the %, leave just the scoring cut-points.

Lines 150 and 155: Is this the Cronbach Alpha obtained in this study? If so, please, state it in the manuscript.

Line 153: Reference 17 is not from the same authors than the present manuscript, and the text made that impression.

Line 179: review English grammar.

Line 190: the awareness sessions should be described in detail earlier in the methods section.

Results

Table 2: change No for N.

Eliminate the parenthesis from the tables and figures captions. For example, Figure 1, instead of Figure (1).

Table 4: Do these variables refer to before or after the awareness sessions? Please, clarify.

Discussion

The discussion needs more references. I do not think it is appropriate to repeat the references mentioned in the introduction. You base your discussion in only 8 references different form the introduction.

Line 247: Please, provide a reference for this demand.

Lines 276 to 285. Please, expand this argument, the link between psychological distress and awareness of inclusive leadership is not enough explained.

Add the limitations for the present study.

Conclusions

The conclusions need to be rewritten. Now, it is a summary of the results. It should be the ideas supported by these results, the application for practice, the suggested lines for future investigations…

Author Response

Dear Respected Reviewer 

Thanks a lot for your great efforts & care

I revised & edited the manuscript. I revised all your points & responded to them accordingly.

I hope it will be the required modifications.

Thank you for your consideration. I look forward to hearing from you.

Reviewer 2 Report

The manuscript is fascinating and very important. It is also from an under-represented social context! It will be eligible for publication after corrections. Thus, I commend the authors' efforts. Consequently, I recommend a revision, and I offer my suggestions based on the following issues, namely (a) contextualization – introduction, (b) contextualization – discussion & limitations, (c) unsupported assertions, and (d) conclusion is too short. See the attachment for details.

Reviewer 3 Report

Dear authors, 

Congratulations to the authors for the presentation of this study. 
It is a very interesting object of study for the scientific community, however, the paper needs further revision on the following points:

  1. Correctly add the reference on line 134.
  2. The presentation of the evaluation tools is open to multiple interpretations. 
    I suggest the presentation of the chosen tools in tables describing their parts, dimensions, scoring scale, items, etc.
  3. I suggest the organization of the results section, with differentiated tables, descriptive legends, explanations of the most important results, bar charts for quantitative data and sectors for %. 
    The current presentation of the results is chaotic.
  4. Review the bibliographic references. 

Kind regards, 

Reviewer 4 Report

The study focuses on the effect of nurse manager inclusive leadership style on nurses' psychological distress during COVID-19 pandemic 14 in Egyptian hospitals. The study is mostly explorative, but I consider it interesting anyway.

There are several suggestions I do have to make the study and article stronger:

  1. Please use the word participants instead of subjects
  2. Please explain the rationale for testing only nurses and not other staff (doctors, for example)
  3. I would like to know more about the leadership questionnaire and why the authors find the construct important
  4. Scoring system does not need to be as bullet points (1.7.1)
  5. Not sure I have found an IRB number
  6. Figure 1. Please avoid pie plot and use histograms instead, which are more informative. In any case figure 1, it can go in the text, no need of a plot for such basic information
  7. Table 4 can go into the text, no need of a table with 1 line
  8. "Recommendations" come after "conclusions" (level 5) but are marked as level 2. Also avoid bullet points

Reviewer 5 Report

This is an interesting study and it is within the remit of the Journal, but it should be deeply revised before it can be accepted for publication.

I have made some recommendations/comments below:

-Introduction is long-winded and repetitive, I suggest to completely eliminate the "significance of the study" section (from lines 74 to 89) as this paragraph repeats elements already previously reported and also anticipates part of the objectives.

-line 33 I suggest to eliminate the word "novel" coronavirus since are more than 2 years the world is facing the pandemic...probably recent may be a better word

-I suggest to eliminate the name of the paragraph relating to the hypothesis and to report directly the hypothesis under the objective. Furthermore, the Authors should clarify the reason for their hypothesis especially for the first one.

-Subjects: first group of participants, the Authors should report how many nurse manager have been excluded, if there were drop-out and or refusals to participate. Also for the second group these information should be provided.

-Paragraph 1.7 should be renamed in MEASURES. When reporting the tool's author please report the first author's name and after the reference number. Furthermore in this section I suggest to eliminate the bold style and the lists that are not necessary. Furthermore as regards measures I suggest to clarify who filled them, the first group of participants, the second or both.

-As regards the pilot study it appeared surprisingly, I suggest to insert it in the aim of the study or in procedure section or to eliminate this paragraph since it does not add relevant information for the study

-paragraph 1.9 and 1.10 should be unified and named PROCEDURE. Plese clearly report the period of data collectio...from-to. I also suggest to better explain the procedure of data collection.

-discussion/conclusion, I encourage to insert a limits section.

Minor: trough the manuscript there are some typos as COVID-19 that is written in different way COVI-19 COVID19 COVID-19; when reporting percentage sometimes there are 3 number after the point other two...

Round 2

Reviewer 2 Report

Thank you. I believe that you have significantly improved your article following my previous suggestion, and therefore the paper warrants publication - after a minor correction.

Minor correction

A language expert must copy-edit and proofread the manuscript to improve correctness, readability, and clarity.

Author Response

Thanks a lot for your great efforts & care

Best Regards,

Reviewer 3 Report

Dear authors, 

I appreciate the incorporation of the suggested modifications. 

The modifications have improved the quality of the paper and clarified its content. 

Congratulations !

Kind regards, 

Author Response

(The authors gave the same response as above.)

Reviewer 4 Report

Nice article

Author Response

(The authors gave the same response as above.)

Reviewer 5 Report

Dear Authors,

I appreciated the revision done but some points that I have highlighted have not be considered. I suggest to:

1) clarify the reasons of your hypothesis, specifically why did the Authors expet " that most of nurse managers will lack knowledge about inclusive leadership"

2) the second hypothesis seems a result and not a hypothesis, please reworded.

3) in the response to reviewer the Authors reported the number of participants excluded in each group according to inclusion/exclusion criteria but this information was not added in the manuscript so I suggest to insert this information

4) as regards measures I suggest to clarify who filled each of them, the first group of participants, the second or both.

5) limits should be reported at the end of discussion before the conclusions and should be discussed and explained to the readers. 

Author Response

(The authors gave the same response as above.)
